# GPR Application on Geothermal Studies: The Case Study of the Thermal Baths of San Xusto (Pontevedra, Spain)

**Mercedes Solla** [1,*] , **Cristina Sáez Blázquez** [2,3] , **Ignacio Martín Nieto** [3] , **Juan Luis Rodríguez** [1] and **Miguel Ángel Maté-González** [3,4]

1. CINTECX, GeoTECH Research Group, Universidade de Vigo, 36310 Vigo, Spain; jlsomoza@uvigo.es
2. Department of Electric, System and Automatic Engineering, Universidad de León, 24004 León, Spain; csaeb@unileon.es
3. Department of Cartographic and Land Engineering, Higher Polytechnic School of Avila, University of Salamanca, Hornos Caleros 50, 05003 Avila, Spain; nachomartin@usal.es (I.M.N.); miguelangel.mate@upm.es (M.Á.M.-G.)
4. Department of Topographic and Cartography Engineering, Higher Technical School of Engineers in Topography, Geodesy and Cartography, Technical University of Madrid, Mercator 2, 28031 Madrid, Spain
* Correspondence: merchisolla@uvigo.es

**Abstract:** Geophysical studies are frequently used on the geothermal field to define and characterize deep structures. However, shallow investigations are also needed for understanding the origin and local potential structures of a promising geothermal site. In this research, it is intended to present a review of the possibilities of the application of ground-penetrating radar (GPR) on the study of geothermal resources and how this geophysical technique can contribute to improving the energy use of these thermal resources. For this, the specific case of application to the investigation of the thermal baths of San Xusto (Pontevedra, Spain) is included in this work, whose interest for the region makes it necessary to perform an in-depth analysis of the original thermal structure. A GPR survey with frequency antennas of 200 and 500 MHz was conducted. Additionally, chemical analyses were performed to characterize the thermal water in the San Xusto site. As a result, a hot spring was detected by identifying reverberation phenomena in GPR imaging due to the presence of metal compounds and silica. Locating the origin of the thermal springs could allow for a more efficient use of the thermal resources as well as the hydrothermal possibilities of the area.

**Keywords:** geothermal reservoirs; near-surface imaging; ground-penetrating radar; geophysical prospecting; thermal structures

## 1. Introduction

Of all the phenomena directly observable on the surface that reveal thermal activity in the underlying geological structure, the emergence of hot springs is one of the most common and yet least studied from a morphological point of view. This may be because, in most cases, the uses of the fluid do not require an exhaustive search for the upwelling zone in the bedrock.

On the other hand, the application of geophysical techniques for the characterization of geothermal resources has become one of the most rewarding branches of knowledge generation in the field of this renewable energy in recent times. There are numerous works that have achieved significant advances in the characterization of geothermal reservoirs from very different technical perspectives. We can cite the works on electrical tomography [1,2], seismic refraction [3], time-domain electromagnetic methods (TDEM), and magnetotelluric [4,5], among others. However, what these works have in common is their focus on large reservoirs, characterizations of thermal properties, or the location of potentially exploitable geological structures at medium/high depths.

In the case of hydrothermal surging for geothermal energy and other uses, hydrothermal fluid surges from families of fractures in the rock. On many occasions, this occurs at a short distance from the surface, in the contact of the bedrock with the sedimentary/altered material that forms the soil. This turned out to be the case of the hydrothermal reservoir studied in this work. Therefore, the correct characterization of the geometry of the fractures with hydrothermal activity, as well as their spatial distribution, can be key when proposing a more direct use of the thermal resource. Given the very shallow position of these springs on many occasions, the normally used geophysical prospecting resources will not be applicable, and it will be necessary to adapt other geophysical techniques not traditionally used in the sector. As will be seen, the ground-penetrating radar (GPR) technique can be adapted to this type of characterization that requires the identification of structures in the subsoil, as well as the provision of information on the possible composition detected in the anomalies.

GPR is a geophysical imaging technique widely used for subsurface exploration and monitoring, due to its high-resolution and continuous imaging [6]. It uses low and high frequencies (from 10 MHz to 6 GHz) for the analysis of the propagation capacity of electromagnetic waves through media with different dielectric constants, thus detecting discontinuities in the subsoil at the interface between adjacent media with sufficient dielectric contrast. The GPR method allows interpreting features of interest, such as stratigraphy and layering, water bodies, cavities, etc.

In [7], GPR was used as a tool for mapping the near-surface structure of the Chingshui geothermal field in Taiwan. The results provided an accurate subsurface image of the area. In [8], GPR was applied to the characterization of the shallow subsurface (0–5 m depth) of the Old Faithful geyser event at the Yellowstone National Park, USA. GPR images revealed different micro-fractures at a 0–5 m depth that fill and drain repetitively, immediately after an eruption and prior to the main eruptive event. In [9], a GPR survey was undertaken over the southern section of the Armstrong Reserve, located in Taupo, New Zealand, to image the shallow subsurface, revealing three distinctive rock types in the subsurface. In [10], the GPR method was used to detect the presence of hot water in the Blawan geothermal field (Indonesia). The underground seepage of hot water was successfully identified. In [11], multi-fold ground-penetrating radar (MFGPR) was applied for extracting information about cross-property relations of interest in the characterization and monitoring of deep geothermal resources. This method was also used as a structural and physical properties validation tool, providing ultra-high-resolution images of the rock volume as well as networks of joints and fractures.

In accordance with the above, there are many more studies that have successfully implemented this geophysical technique in the geothermal field, for the imaging of hot spring deposits [12], underlying hydrothermal alteration and hot spring vents [13], detection of fractures [14,15], evaluation of hot dry rock resources [16], or even the determination of geothermal properties in a shallow system [17].

Regarding the case study presented in this work, the area under investigation turned out to be a location with hot springs known in the region for a long time. However, given the thickness of the sedimentary layer on the bedrock, the exact location of the families of fractures through which the hydrothermal fluid emerges were unknown. Through the application of GPR, an attempt will be made to establish the position of the specific springs on the ground. This will be undertaken by detecting metal precipitation compounds in the areas of direct upwelling of the fractures towards the upper sedimentary/altered material. It should be highlighted here that the GPR signal reverberates (known as "ringing signal") with the presence of metal [18]. However, the detection of hot springs by identifying reverberation phenomena is not adequately addressed in the published literature. Using available chemical analysis of the hydrothermal fluid, it can be inferred that there are real possibilities of detecting such accumulations given the concentrations of some metals found in the chemical reports. All this information will, in turn, contribute to a more efficient use of the thermal resource, exploiting and maximizing the hydrothermal possibilities of the

area, also considering the existence of hydrothermal evidence in many parts of the Spanish territory [19].

Table 1 compiles the most related works in the literature, including information about the reflection pattern used to detect hot springs and the frequency antennas used.

**Table 1.** GPR signatures generally associated with hot springs.

| Application | GPR Signatures | Frequency Antennas | Reference |
|---|---|---|---|
| To image a buried sinter (hot springs that discharged silica-rich, alkali-chloride water) | More prominent and stronger reflections dominantly horizontal produced by unaltered siliceous sinter. | 200 MHz | [9] |
| To locate buried siliceous sinters (hot spring rocks) and other thermal features. | Hot spring water discharges and cools to below 100 °C, silica precipitates and accumulates to form a sinter deposit: (i) in settings where the sinter deposit is thick, a series of strong, flat-lying reflections were displayed; (ii) over time, the alkali-chloride may change to acid sulphate and breakdown of the sinter into clay, thus resulting into weak flat-lying reflections. | 200 and 400 MHz | [12] |

## 2. Description of the Study Site

The Baths of San Xusto are located on the left bank of the Lérez River, in the municipality of Cerdedo-Cotobade (Pontevedra, Spain). In Figure 1, it is possible to observe the exact location of the site.

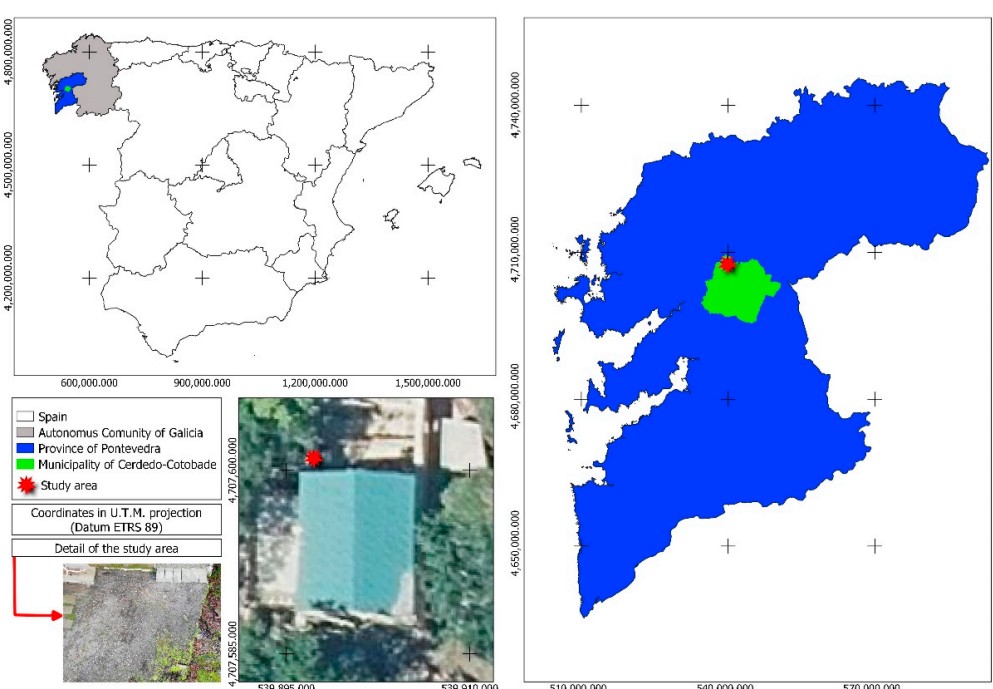

**Figure 1.** Location of the thermal baths of San Xusto in Pontevedra, NW Spain.

It is a hot spring that was used by the residents of the area since time immemorial thanks to the existence of several springs of this mineral water in the area.

In the mid-19th century, the Cotobade council built three stone basins that came to satisfy the demand of 400 water users per year. In 1889, the residents of Cotobade had the possibility of using the waters by only paying a previously agreed municipal tax.

At the beginning of the 20th century, different improvement works were carried out, such as the construction of a 4 m-high wall to separate the baths from the river, gathering the flows in a single source for subsequent conduction to a new reservoir and the construction

of a shed with sinks for bathrooms. It was also agreed that one of the municipal doctors would establish his residence in San Xusto to meet the demands of the water users.

The original spa, a large two-story building next to the river, was built in 1914 and remained in service until 1934, when the decay of the baths in Galicia began.

Near these buildings, there is another much older one that could be related to the first construction in relation to the hot springs. On the bed of the river Lérez and in the location where the upwellings emanate, the placement of stones and rocks can be seen, used to dam the waters as much as possible for use on the river. It is possible to appreciate the springs observing the characteristic white color and the peculiar smell of this type of mineralized water.

Next to the river, the remains of several mills from the end of the 19th century can also be seen, which together with the spa, provide an idea of the activity of the area in those times.

### 2.1. Geological Characterization

The study area is characterized by the presence of Herculean granitic rocks. A set of granites of two micas of the alkaline type and granodiorites up to diorites of the chalco-alkaline type are grouped under this term. Within the complex, it is possible to recognize a wide range of two-mica granites based on small differences in, for example, granularity, presence, or absence of mega crystals, type and size of crystals, degree of deformation, foliation, and color index. It is also common to find intrusive relationships between varieties, constituting very intimate mixtures that are generally not mappable at these scales.

The granite formations observed in the area show signs of deformation: on a macroscopic scale, a slight phyllonitization is sometimes noted, and microscopically they show quartz, fractured feldspars, and curved micas.

Figure 2 shows a detailed description of the geology environment around the baths of San Xusto. As can be observed, the contacts with the metasedimentary rocks are always very clear, with the presence of andalusite or sillimanite at various points of contact. Figure 2 also highlights the presence of faults and fractures, as well as subvertical foliation, which could explain the origin and circulation of the emergence of mineral waters in the site.

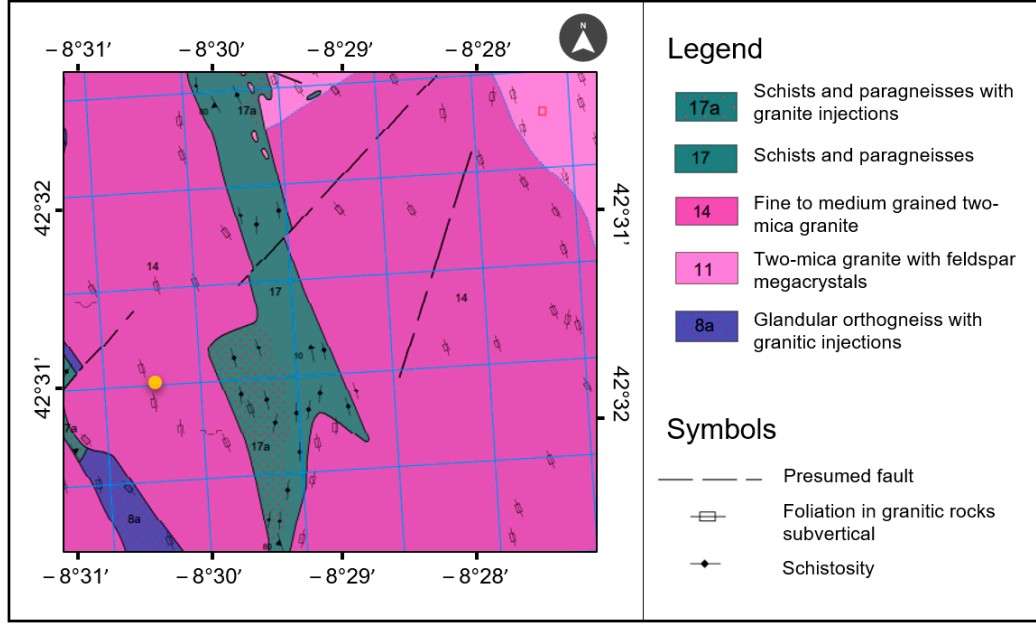

**Figure 2.** Geological characterization of the area under study. Coordinates are presented as latitude and longitude.

### 2.2. Resource Origin

San Xusto waters are very weakly mineralized, sodium bicarbonated, fluoridated, sulphurated, and with a temperature of 21 °C. As for the pH of the mineral waters, due

to the hydrolysis of the silicates, its value is around 9. The properties attributed to these waters are traditionally related to diseases of the skin, liver, rheumatism, and arthritis.

Nowadays, an active project includes the rehabilitation of the fountains and the water tank, which made it possible to discover the existence of thermal waters in the area, or the reconstruction of three old mills, one of which was transformed into a sauna. The miller's house, for its part, was rebuilt to house the biomass boiler warehouse with which the spa will work, and the accesses were improved with new parking areas and the conditioning of the entire area. On top of the above, the regularization file for these mineral-medicinal waters, which were not registered, was prepared during the last year, consisting of twelve analyses over the year.

Despite the importance and role of the mentioned baths of San Xusto in the region, the origin of these mineral waters is not clear. These kinds of resources do not normally originate in the subsoil, but rather seep from the top of a valley, and on their way down they cross areas of high concentration of various minerals that confer their characteristics. The lack of evident information about the site has led to an in-depth study of the possible fractures that gave rise to the emergence of the waters currently found in the site.

Table 2 shows the results of the laboratory measurements performed on the waters collected in the San Xusto site. Table 2 only presents the most representative values that could have an influence on its detection using GPR techniques.

**Table 2.** Results of the laboratory measurements carried out on the waters of the site.

| Parameter | Test Method | Limit * | Results | Units |
|---|---|---|---|---|
| Boron | ICP-MS/002-a | 0.010 | 0.14 | mg/L |
| Fluorides | CI/002-a | 0.015 | 12 | mg/L |
| Silica | ICP/014-a | 0.26 | 56 | mg/L |
| Ammonium | COL/007-a | 0.050 | 0.27 | mg/L |
| Selenium | ICP-MS/002-a | 0.3 | 0.5 | μg/L |
| Aluminum | ICP-MS/002-a | 10 | 50 | μg/L |
| Iron | ICP-MS/002-a | 5 | 27 | μg/L |
| Electrical Conductivity (20 °C) | EL/001-a | 10 | 297 | μS/cm |
| Dry residue (180 °C) | GRV/006-a | 30 | 248 | mg/L |
| Dry residue (260 °C) | GRV/006-a | 30 | 235 | mg/L |
| pH | EL/002-a | - | 8.8 | pH unit |

* Concentrations considered usual in waters of the area with the possibility of being destined for human consumption.

## 3. GPR Data Acquisition and Processing

### 3.1. GPR Field Survey

GPR data acquisition was conducted with a ProEx system from the company Malå Geoscience, equipped with two ground-coupled antennas having central frequencies of 500 and 200 MHz. Data acquisitions were carried out using the common-offset mode by distance, and the acquisition parameters were a trace-interval of 1 cm and a total time window of 83 ns (composed of 576 samples per trace by default) for the 500 MHz antenna; whereas for the 200 MHz antenna, these parameters were a trace-interval of 5 cm and a total time window of 200 ns (composed of 592 samples per trace by default). For data acquisition, both the shielded and unshielded antennas, 500 and 200 MHz, respectively, were mounted in their respective survey carts with an odometer wheel to measure the profile length and to control the trace-interval distance (Figure 3c). The raw data collected in this study are openly available in the GitHub repository [20].

A three-dimensional (3D) GPR methodology was carried out, and a total of 26 parallel 2D profile lines were registered (for each antenna) at regular intervals of 20 cm spacing. It should be noted that the profile lines were conducted only in one direction (*x*-line). Figure 3a represents the GPR grid (3.5 × 5 m), with the position of the origin (0,0) and the direction for both *x*-line and *y*-line increments.

As seen in Figure 3b, the existence of hot springs through the study area is confirmed by the presence of a thermal spring coming out from the retaining wall in the study area at a depth of ~3.0 m.

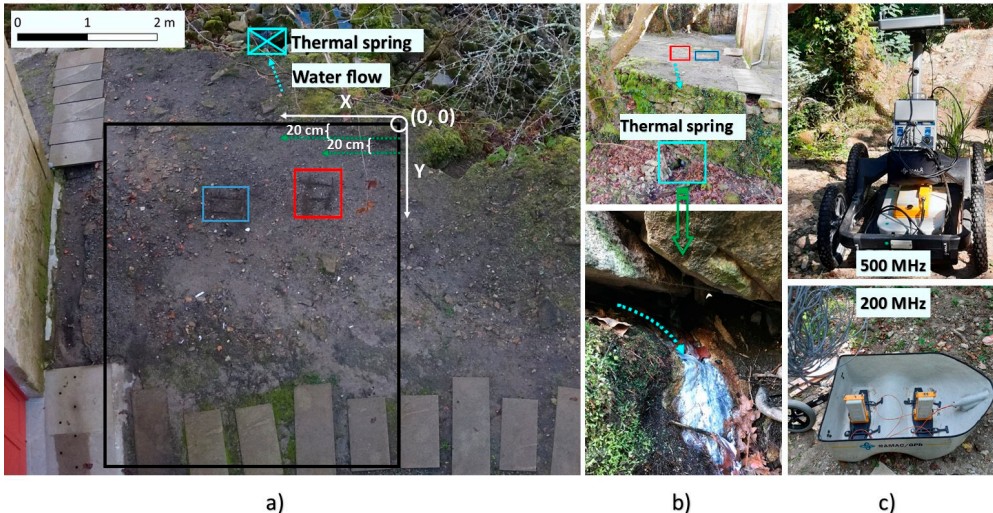

**Figure 3.** GPR setup: (**a**) 3D grid showing the location of the anomalies identified (red and blue boxes), (**b**) existence of a thermal spring in the area, and (**c**) 500 and 200 MHz antennas used for data collection.

### 3.2. GPR Data Processing

The GPR signals received were processed with the ReflexW software [21] by applying the processing sequence described in Table 3. The objective was to correct the down-shifting of the radar section due to the air–ground interface (time-zero correction), to remove the initial DC signal component, or DC bias, in the radar traces as the averaged level of the signal is moved from zero amplitude to a different value (dewow), to amplify the received signal to mitigate the energy decay (gain), to remove horizontal continuous low-frequency reflectors (subtracting average), to remove both low- and high-frequency signal noise in the vertical and horizontal directions (bandpass), and to suppress strong clutter (migration). The velocity used for migration was previously estimated by hyperbola fitting.

**Table 3.** Filters and settings used for GPR data processing.

| Filters | Parameters | |
|---|---|---|
| | **500 MHz** | **200 MHz** |
| Time-zero correction | −6.4 ns | −20.0 ns |
| Subtract-mean (dewow) | Time window: 2 ns | Time window: 5 ns |
| Gain function | Linear: 2 and Exponential: 2 | Linear: 1 and Exponential: 1 |
| Subtracting average | 250 traces | 250 traces |
| Bandpass (Butterworth) | Lower: 300 and Upper: 950 MHz | Lower: 100 and Upper: 230 MHz |
| Migration (Kirchhoff) | Velocity: 0.1 m/ns | Velocity: 0.1 m/ns |

These processed data were then exported to the 3D data interpretation module of the same processing software for the 3D cube creation. In addition to the equidistant profile increment, all the radargrams acquired with each antenna frequency were registered with equal trace-distance and time increment (vertical sampling), so the cube was generated without interpolation. This 3D data allows for the extraction of either parallel or crossing 2D lines to enhance the spatial correlations of the reflectors.

## 4. Results and Discussion

From the GPR data produced with the 500 MHz antenna, shown in Figure 4, the presence of two anomalies characterized by stronger amplitude values is easily observed,

showing reflection patterns in the form of signal reverberation. These reflections are seen to be extended from ~1.0 to 1.5 m in y-line and 0.5 to 1.0 m in *x*-line, at a time-depth of ~10 ns (red boxes), and from ~1.2 to 1.6 m in y-line and 1.8 to 2.2 m in *x*-line, at a time-depth of ~15 ns (blue boxes).

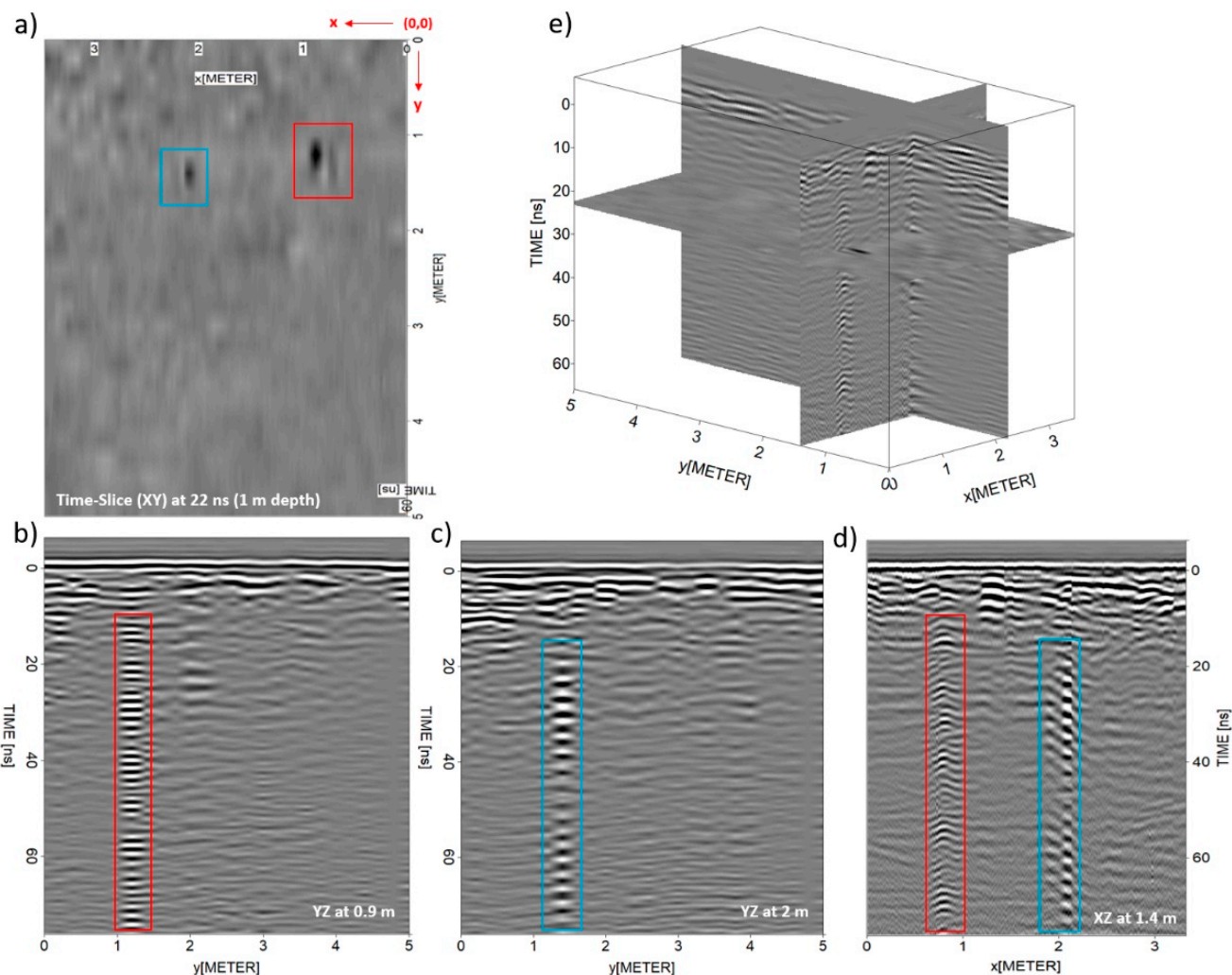

**Figure 4.** 500 MHz data: time-slice or XY image at 1 m depth (**a**), YZ image at 0.9 m in *x*-line (**b**), YZ at 2.0 m in *x*-line (**c**), XZ or conventional radargram at 1.4 m in *y*-line (**d**), and 3D perspective (**e**).

Regarding the GPR data produced with the 200 MHz antenna, presented in Figure 5, only one anomaly was identified, which is also characterized by stronger amplitude and signal reverberation. Moreover, this reflection is seen to be extended from ~1.4 to 1.8 m in *y*-line and 1.5 to 2.3 m in *x*-line, at a time-depth of ~50 ns (blue boxes). It is important to mention here that the dimensions of the survey carts (see Figure 3c) differ, resulting in the difference obtained in the *y*-line and *x*-line positioning. Taking the (0, 0) coordinates of the grid (Figure 3a) as a reference, the real coordinates (*x,y*) for the initial mid-point of the Tx and Rx antennas were (0.5, 0.3) and (0.4, 0.5) for the 500 and 200 MHz, respectively.

Observing the reflection pattern attributed to the geothermal sources interpreted in Figures 4 and 5, it consists of strong flat-lying reflections showing reverberation in the signal pattern, which is typically caused by metal elements. The explanation for this may lie in the accumulation of metal compounds and silica due to precipitation [22] in the interface between the end of the fractures in the granite rock, through which the thermal water flows, with the sedimentary material that forms the soil. This accumulation of metals, resulting from the change in the physical conditions of the hydrothermal fluid, would

represent a small hydrothermal proto-field that is liable to be detected by the GPR, thus accurately locating the thermal surging within the family of rock fractures. This hypothesis is reinforced by the analytical results presented in Table 2, where the existence of metal salts in significant concentrations can be seen. It is also worth mentioning the presence of Boron, which, together with the concentrations of the other compounds, makes us think that we could be facing a hydrothermal phenomenon similar to that of Lardarello, Italy, where these precipitations of metal salts in the fluid eruption spots are common [23]. Given the composition of the hot springs found in the chemical analysis and the correspondence with other aforementioned hydrothermal upwelling areas, the predominant substances in the upwelling spots detected must be iron and silica salts, accompanied by other metal precipitations such as aluminum salts, boron, etc.

The signal behavior observed in this case study is in agreement with the results reported throughout the literature, especially in those with similar purposes [9,12].

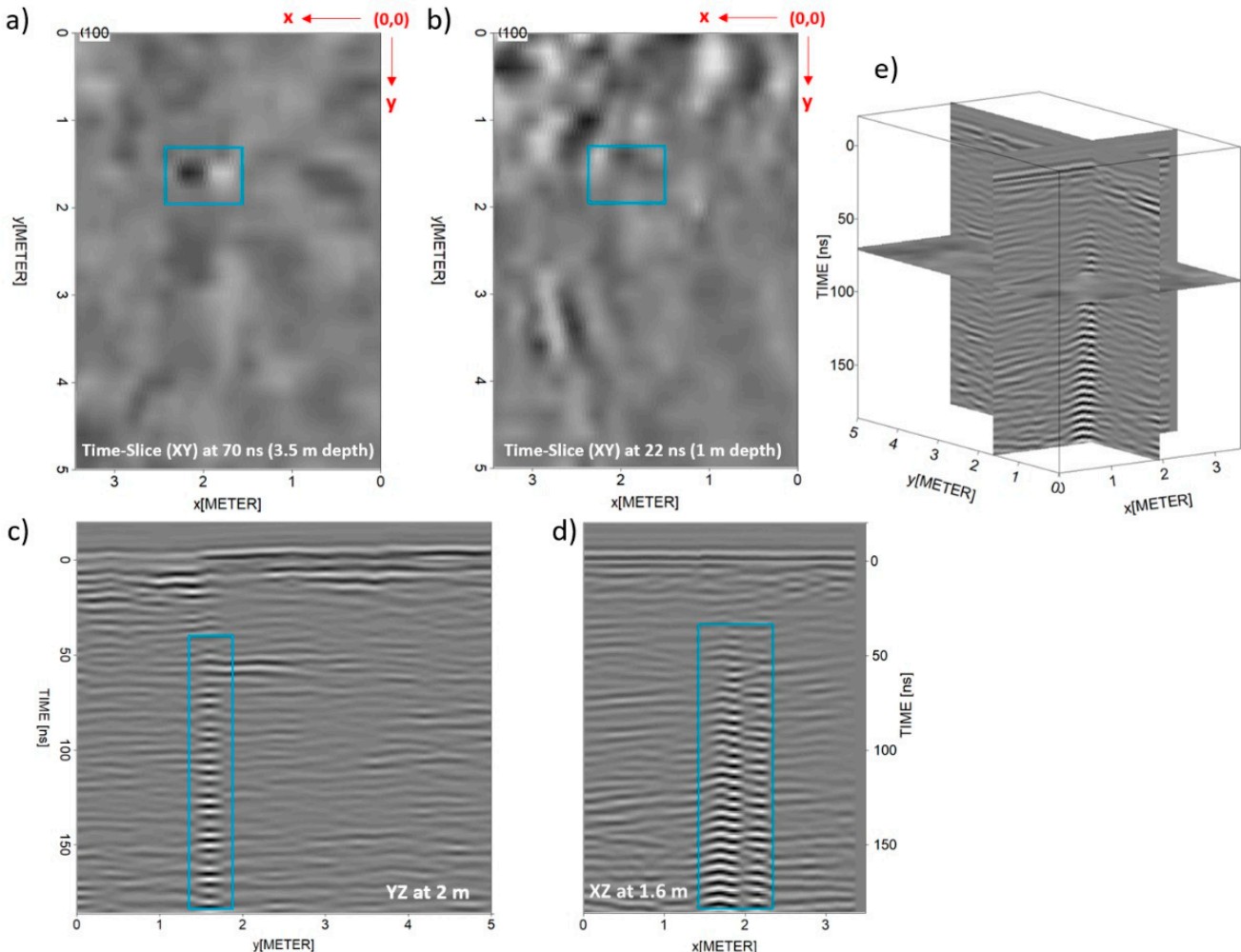

**Figure 5.** 200 MHz data: time-slice or XY image at 3.5 m depth (**a**), time-slice at 1 m depth (**b**), YZ image at 2 m in *x*-line (**c**), XZ or conventional radargram at 1.6 m in *y*-line (**d**), and 3D perspective (**e**).

To estimate the depth of the anomalies interpreted as the thermal surging, the travel-time distance (ns) axis of the GPR images was transformed into a depth (m) axis using a radar-wave velocity of 10 cm/ns, as previously estimated by hyperbola fitting. Regarding the 500 MHz data, the thermal sources were registered at ~10 and ~15 ns (red and blue boxes, respectively), which resulted in ~0.50 and ~0.75 m in depth, respectively. On the other hand, the thermal source interpreted from the 200 MHz data at ~40 ns (blue box) resulted in a ~2.0 m depth. The different depths obtained for the thermal source highlighted in the

blue box (the one detected by the two frequencies used) can be explained by the difference in the resolution range. Lower frequencies provide higher capacity of penetration but lower resolution, while higher frequencies yield lower capacity of penetration but higher resolution. Under optimal conditions, the 500 MHz frequency antennas yielded vertical and horizontal spatial resolutions of 5 and 90 cm (considering a vertical distance of 1 m between the antenna and the reflector surface), respectively, whereas the 200 MHz frequency antennas provided a vertical resolution of 12.5 cm and a horizontal resolution of 150 cm. This could explain why the 500 MHz antenna is detecting two different springs. Observing the 200 MHz data in Figure 5d, it is possible to appreciate a wider footprint of the anomaly than for the 500 MHz data and, what is more, it could be said that two reflectors overlapped. Figure 6 illustrates the simplified GPR antenna footprint for bistatic dipole antennas [24]. Considering the equation in Figure 6, the long dimension radius of footprint (A) for the 500 and 200 MHz (with vertical distance of 1 m between the antenna and the reflector surface) resulted in 40 and 48 cm, respectively. This ratifies how a lower-resolution image would cause features to appear larger than they are. It should also be mentioned that higher-frequency antennas, such as the 500 MHz antenna, are more sensitive to ambient noise, such as the presence of metal compounds from the water.

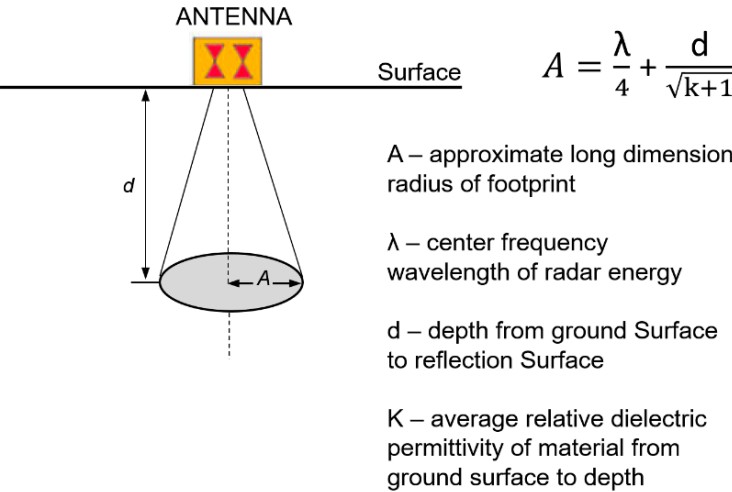

$$A = \frac{\lambda}{4} + \frac{d}{\sqrt{k+1}}$$

A – approximate long dimension radius of footprint

λ – center frequency wavelength of radar energy

d – depth from ground Surface to reflection Surface

K – average relative dielectric permittivity of material from ground surface to depth

**Figure 6.** Simplified GPR antenna footprint (Fresnel zone) for bistatic dipole antennas.

Moreover, Figure 7 presents different XZ and XY images from the 200 MHz 3D cube, highlighting: (i) the presence of a more prominent reflection that could be produced by a fracture in the rock (dotted orange lines) and (ii) the footprint of this anomaly interpreted as a fractured zone, showing the spatial correlation with the springs' surging detected. However, it should be mentioned here the possibility that this reflection can be a corner reflection (or aerial) from the building walls. To clarify this, an additional 200 MHz profile line was acquired in the transversal direction along this area, further away from the building façade. As shown in Figure 8a, a stronger reflection was identified at the area previously interpreted as fractured. What is more, Figure 8b shows the 500 MHz XZ images analogous to the XZ images in Figure 7b,c, which ratifies the presence of a prominent stronger reflection at ~40–50 ns in depth. All these results seem to agree that more than likely, there is a fracture in that area.

It is therefore concluded that GPR allowed to detect two thermal sources in the study area, although only one spring arises from the retaining wall. This could have two different interpretations: (i) the two sources joined in their trajectory, or (ii) only one of the sources is currently active.

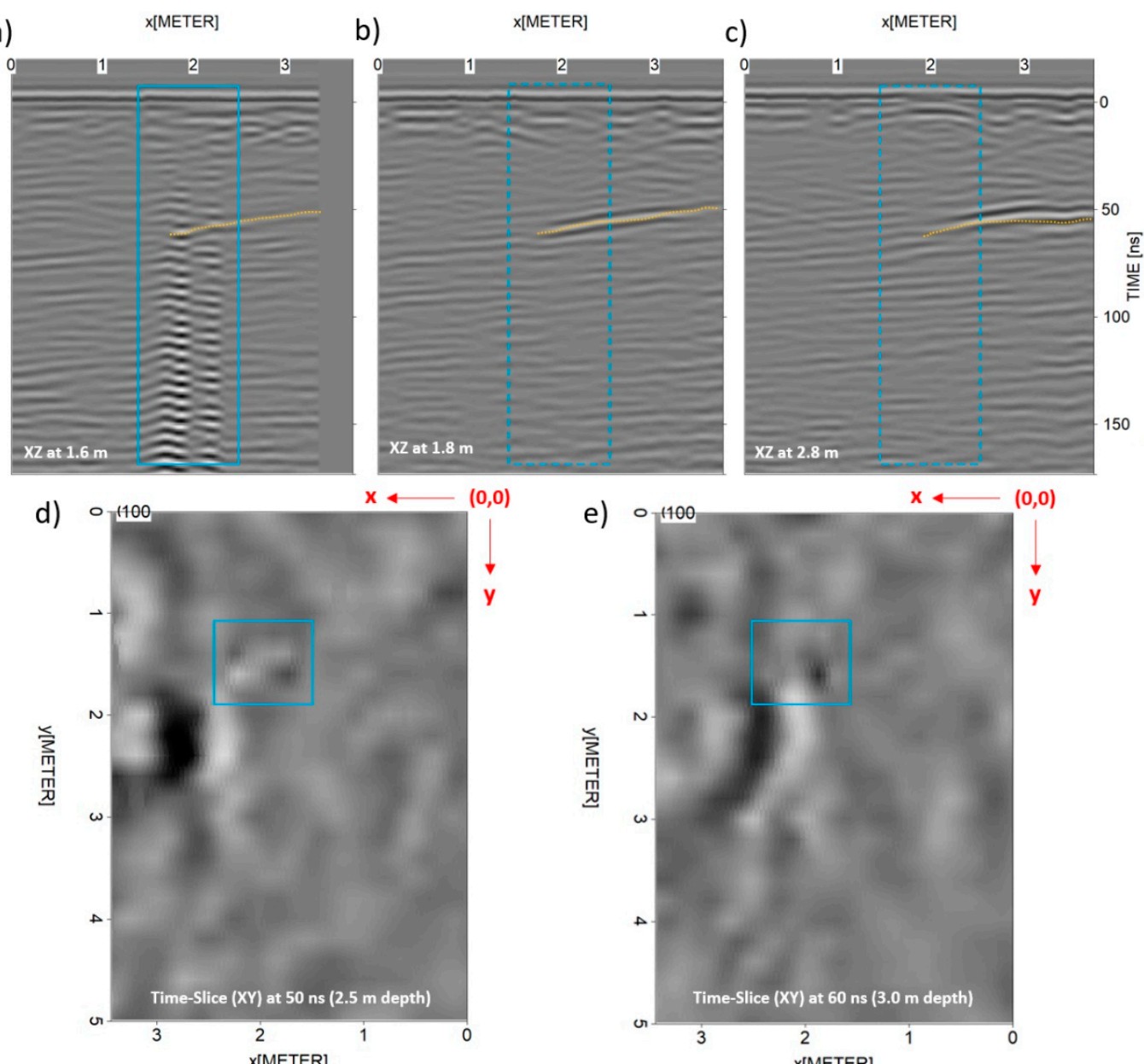

**Figure 7.** 200 MHz data showing the spatial correlation between the springs' surging and a probable fractured zone detected: XZ image at 1.6 m in *y*-line (**a**), XZ image at 1.8 m in *y*-line (**b**), XZ image at 2.8 m in *y*-line (**c**), time-slice or XY image at 2.5 m depth (**d**), and time-slice or XY image at 3.0 m depth (**e**).

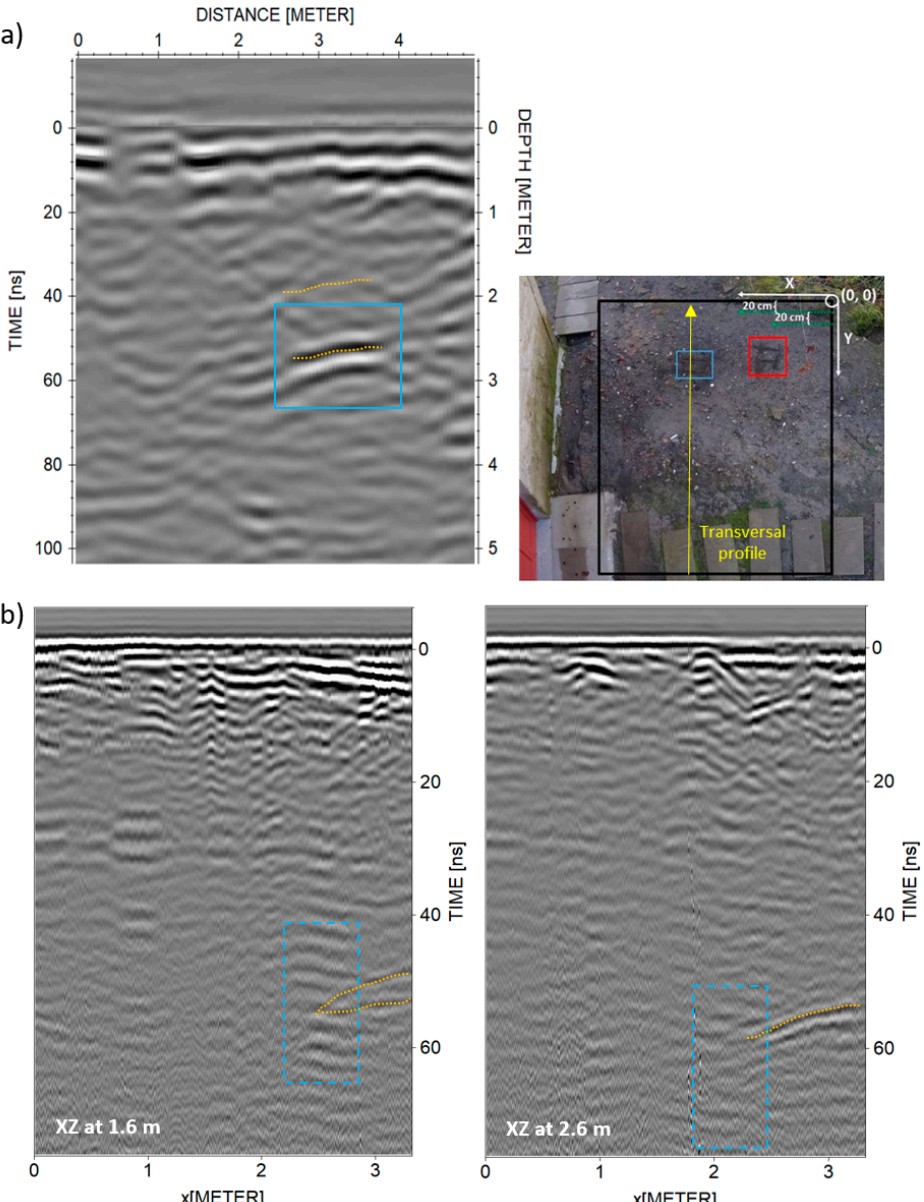

**Figure 8.** Additional 200 MHz transversal profile acquired along the probable fractured area (**a**), and 500 MHz XZ at 1.6 and 2.6 m in *y*-line (**b**).

## 5. Conclusions and Future Research Perspectives

The case under study shows the application of GPR to an in-depth study of thermal resources. The interpretation of the geological information of the site, the geophysical results, as well as the mineral composition of the waters has allowed to identify the original structure that possibly gave rise to the current hydrothermal and mineral source. The combination of all sources of information results in important knowledge about the most probable origin of the thermal spring and its evolution in time and depth. It also offers the possibility of establishing relationships in terms of hydrothermal behavior with other well-known deposits (Larderello, Italy). This connection between similar hydrothermal springs could serve to apply successful experiences in this type of deposit carried out in other places.

The GPR method allowed to identify two different thermal sources, in the form of strong flat-lying reflections showing signal reverberation, in the area under study. In this regard, the higher resolution of the 500 MHz data allowed distinguishing between the two reflectors, whereas in the 200 MHz data, these reflectors seemed overlapped. In

addition, the 3D data (time-slices) enabled for better knowledge of their origin and extent. Accordingly, it was possible to provide an estimation of the depth of the springs' surging. For the latter, the 200 MHz data appear to provide a more reliable depth since, although this frequency antenna has a lower resolution, it is less affected by ambient noise and ringing noise from metal components in the thermal water. Furthermore, this frequency allowed to identify a probable fractured zone that could be the origin of the springs' surging.

The results of this study directly contribute to the revaluation of the area through a greater knowledge of the resource and its origin. In turn, the interpretations performed in this research could also be valuable in the current rehabilitation initiatives that are being carried out in the San Xusto baths. The interest in the site, and hence the number of potential visitors, especially for those out of the region, who may find the study an opportunity to visit in situ the origin of this kind of deposit, are expected to increase.

Further research will include the use of complementary geophysical methods, such as the electrical resistivity tomography (ERT), that could provide additional information on the presence of fractures and water bodies. Additionally, and given that the interpretation of the position of the specific springs on the ground is an indirect observation (which assumes direct upwelling of the fractures towards upper material), future research will be focused on direct sampling aimed at verifying the conclusions and main results of the present work. Moreover, it is also expected to confirm that the detection of the points of emergence in the fractures by the GPR is determined by the accumulation of silica, iron compounds, and other elements, which most certainly produce signal reverberation. Sampling through soundings in the area will be proposed as one of the possibilities to access the alleged upwelling areas identified in the radar images produced. All these results will allow to verify if it is worth reconditioning the site to collect the thermal water directly from the granite rock.

**Author Contributions:** Conceptualization, all authors; GPR data acquisition and processing, M.S. and J.L.R.; geothermal characterization and thermal analyses, C.S.B., I.M.N. and M.Á.M.-G.; writing—original draft preparation, review and editing, all authors; project administration, M.S. All authors have read and agreed to the published version of the manuscript.

**Funding:** This work has received funding from the Xunta de Galicia—GAIN—through the project ENDITí (Ref. ED431F 2021/08). M. Solla acknowledges the grant RYC2019–026604–I funded by MCIN/AEI/10.13039/501100011033 and by "ESF Investing in your future". The authors also want to mention the regional project SA102P20 funded by the Junta de Castilla y León and FEDER European funding.

**Data Availability Statement:** The data presented in this study are openly available at https://github.com/MerchiSolla/GPR_Geothermal/tree/raw (accessed on 9 March 2022) under a GNU General Public License (v3) and archived at DOI: https://doi.org/10.5281/zenodo.6341953.

**Acknowledgments:** The authors want to thank the City Council of CERDEDO-COTOBADE for all the technical support and information provided to the project.

**Conflicts of Interest:** The authors declare no conflict of interest. The funders had no role in the design of the study; in the collection, analyses, or interpretation of data; in the writing of the manuscript, or in the decision to publish the results.

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
