# Peer review of "GPR Application on Geothermal Studies: The Case Study of the Thermal Baths of San Xusto (Pontevedra, Spain)"

_remotesensing, doi:10.3390/rs14112667_

Round 1
Reviewer 1 Report
The paper presents a new near surface GPR investigation of a thermal spring, targeting recognition and mapping of the features and source point of the spring. The paper presents processed and interpreted radargrams from each antenna and at least one depth/time slice to support their interpretations. The work finds metallic deposits (present in ringing GPR reflections) with two antennas and concludes that these are consistent with the geochemistry of the site and are likely the sources of the thermal springs.
In general, the paper is sound and thorough in its analysis and conclusions. However, some claims are not well supported: the presence of the fractured zone in the 200 MHz scans, the difference in footprint between the two antennas, and the specific contributions of the paper should be improved.
Comments:
- Table 1 outlines many relevant GPR studies, but focuses on amplitude analyses. This paper does not focus on amplitude analyses, rather a combination of chemical and visual analyses. Consider significantly summarizing the information in the table to be more relevant to the work at hand. For example, including a study that uses the metallic ringing/reverberation to identify any kind of geological feature.
- Lines 266-274: Be specific about what metals are precipitating. The manuscript refers to the full chemical analysis in Table 2, but does not discuss which metals are most likely to precipitate in these conditions. Is it Iron and aluminum?
- Figure 6 "Probable fractured zone" or Corner Reflection(?): in the 200 MHz unshielded scans, the paper claims to have found a fractured zone (in yellow). The manuscript must address the possibility that this reflection in the data is the ground wave reflecting off of the building at the end of the transect (sometimes called a corner artifact). This interpretation must be revised, either by additional justification or revising the interpretation of the reflection.
- Concision: a few sections of the manuscript were too long and repetitive and would benefit from revision. Specifically: last 2 paragraphs of introduction (lines 69-87), which also contains a sentence fragment (line 83-84) and unclear sentence (lines 79-82). Section 3.1 contains too much repetition and needs to be edited. The discussion of economic implications in the conclusion, while related to the motivation of the study, is not actually a conclusion that is supported or investigated in the paper and should be shortened or removed.
- Section 2: It is not clear what contributions of this paper are novel. The literature review (section 2) outlines what others have done to investigate geothermal sources and springs, but does not point out where the manuscript falls in this area of research. Is this a new kind of location, or is the contribution an identification of thermal springs in GPR scans by identifying metallic deposits? Similarly, in lines 323-326: Be specific about what "important knowledge" in particular is being contributed.
- Line 297: The difference in size of feature between the two antennas is mentioned. However, there is also a difference in vertical and horizontal spatial resolution between the antennas. The reader must be convinced that the difference is size is not due to interpolation between traces spaced further apart ("trace resolution" of 5 cm for the 200 Mhz and 1 cm for the 500 MHz, Section 4.1), or due to the different vertical (depth) resolution of the two antennas (5 cm for 500 Mhz, and 12.5 cm for 200 MHz). This should be clarified to differentiate this vertical resolution from horizontal (in line trace spacing). The interpretation of the sizes ("footprint") of the spring.
Suggested Clarifications:
Line 132: what is meant by "pylon"? ("Cotobade council built three stone pylons that...")
Lines 157 - 160: delete "etc." from the end of this sentence.
Line 180: What is meant by "thermal implication"? temperature?
Figure 3a: add an approximate scale to indicate total scanned area size.
Grammatical edits:
sentences on lines 36-39, 83-83, needs revision.
Line 115: "more many" should be "many more"
Line 191: "analyzes" should be "analyses" to be the plural of analysis (noun, not verb)
Line 244: "these data were" rather than "this data was"
Author Response
First of all, the authors would like to thank reviewer 1 for the constructive comments provided, which undoubtedly have contributed to improve the quality of this work.
Please, find our responses in the attached file.

Reviewer 2 Report
This paper presents a case study of GPR application to identify the origin structure of the geothermal site in San Xusto, Spain. However, the novelty is not clearly addressed. Specific comments are given below.
- Section 2 should be merged in Introduction.
- Section 3 is lengthy. Please remove unimportant description.
- The meaning of the data in the "Limit" column in Table 2 is unclear, please provide more explanation on it.
- It is recommended to add a detailed description of the metal compounds in the thermal springs, such as its dielectric properties and elemental analysis, and explain its responses in the GPR image.
- The authors claim that the values given in Table 2 could have influence on the GPR detection. Are there any literatures to prove it?
- It is concluded that the two reflectors in the 500 MHz GPR slice overlapped in the 200 MHz slice, due to the degraded resolution. However, two slices in Figs. 5-6 are at different depths, i.e. 1 m and 3.5 m. In order to verify this statement, a GPR slice of the 200 MHz antenna at the depth of 1 m should be presented.
- It is mentioned that the two antennas have different sizes, resulting in a biased coordinate. Please simply resist the two coordinates in two GPR images of different center frequencies, according to the mid-point of the Tx and Rx antennas.
- Many sentences are hard to be read. It is strongly recommended to thoroughly proof the grammar and syntax.
- Some figures are unclear. Please improve the resolution.
Author Response
First of all, the authors would like to thank reviewer 2 for the suggestions provided, which have contributed to improve the quality of this work.
Please, find our responses in the attached file.

Reviewer 3 Report
A case study of GPR application in geothermal studies is represented in this paper. This is interesting topic and, as authors have shown, mostly new area of GPR usage. The paper is well structured, figures and tables are appropriate and graphic quality is fine. References are appropriate, relevant and well cited.
However, there are some drawbacks that should be addressed. Although the paper is well-written in general, it is not always easy to read. Some sentences are too long, some are confusing and some contain incorrect grammar. Several specific comments are pointed to these issues, but it would benefit for the paper to be checked once again. Also, a fracture is detected (as shown in Fig. 6), but it is not investigated any further. For instance, authors do not say if this fracture can be seen in 500MHz 3D image or not and it is not mentioned in the final paragraph of section 5. In my opinion this is interesting finding and authors should discuss this a bit further.
Specific comments:
Line 36-39 – This sentence is too long and not easy to understand. It should be rearranged.
Line 49-52 – This sentence should be split into two: …fractures in the rock. On many occasions this occurs at a short distance… (also occurs, instead of occur).
Line 83-87 – This paragraph should be corrected and rearranged. First sentence is grammatically incorrect. I assume instead of ‘…many parts of Spanish territory [7].’ It should be ‘…many parts of Spanish territory [7], the possibility of applying…’, but then the entire paragraph is one long sentence.
Section 4.1 – Grid dimensions are not given. From Fig. 4 it seems that they are 3.5mx5m, but they should be written in the text as well. Also, profiles were done only in one direction, not both. Authors should explain this briefly.
Line 218 – ‘survey carts’ instead of ‘surveys cart’.
Line 226-227 – I assume 26 profiles for each antenna, i.e. 52 overall. It should be stated clearly.
Line 317-319 – Fig. 6 caption should contain the information on central frequency (although it is written in the text, but it should be here as well).
Author Response
First of all, the authors would like to thank reviewer 3 for the constructive comments provided, which undoubtedly have contributed to improve the quality of this work.
Please, find our responses in the attached file.

Round 2
Reviewer 1 Report
The reviewer wishes to thank the authors for their successful efforts to clarify and improve the manuscript.
Point 3, focusing on the different sizes in footprint of the spring with the two frequencies was not clearly worded. The comment was meant to ask for additional justification to attribute the different sizes of the spring reflection to the antenna resolution or to the actual size of the feature. A lower resolution image would cause features to appear larger than they are. Some additional revision can be made (though the changes made to clarify the vertical and horizontal resolution are also helpful).
